# Interaction between a Novel Oligopeptide Fragment of the Human Neurotrophin Receptor TrkB Ectodomain D5 and the C-Terminal Fragment of Tetanus Neurotoxin

**DOI:** 10.3390/molecules26133988

**Published:** 2021-06-30

**Authors:** Ana Candalija, Thomas Scior, Hans-Richard Rackwitz, Jordan E. Ruiz-Castelan, Ygnacio Martinez-Laguna, José Aguilera

**Affiliations:** 1Molecular Biology Department, Institut de Neruociènces and Biochemistry, Medicine Faculty, Universitat Autònoma de Barcelona (UAB), 08193 Barcelona, Spain; anacandalijaiserte@gmail.com (A.C.); jose.aguilera@uab.cat (J.A.); 2Faculty of Chemical Sciences, BUAP, Puebla 72000, Mexico; toj_cdai90@live.jp (J.E.R.-C.); ignacio.martinez@correo.buap.mx (Y.M.-L.); 3Peptide Specialities Laboratory, Im Neuenheimer Feld, Univerisity Campus, 69120 Heidelberg, Germany; rackwitz@peptid.de; 4Center for Biomedical Research Network on Neurodegenerative Diseases (CIBERNED), 08193 Cerdanyola del Vallès, Spain

**Keywords:** clostridium neurotoxins, neurotrophin receptors, carboxyl-terminal domain of tetanus neurotoxin, brain-derived neurotrophic factor

## Abstract

This article presents experimental evidence and computed molecular models of a potential interaction between receptor domain D5 of TrkB with the carboxyl-terminal domain of tetanus neurotoxin (Hc-TeNT). Computational simulations of a novel small cyclic oligopeptide are designed, synthesized, and tested for possible tetanus neurotoxin-D5 interaction. A hot spot of this protein-protein interaction is identified in analogy to the hitherto known crystal structures of the complex between neurotrophin and D5. Hc-TeNT activates the neurotrophin receptors, as well as its downstream signaling pathways, inducing neuroprotection in different stress cellular models. Based on these premises, we propose the Trk receptor family as potential proteic affinity receptors for TeNT. In vitro, Hc-TeNT binds to a synthetic TrkB-derived peptide and acts similar to an agonist ligand for TrkB, resulting in phosphorylation of the receptor. These properties are weakened by the mutagenesis of three residues of the predicted interaction region in Hc-TeNT. It also competes with Brain-derived neurotrophic factor, a native binder to human TrkB, for the binding to neural membranes, and for uptake in TrkB-positive vesicles. In addition, both molecules are located together in vivo at neuromuscular junctions and in motor neurons.

## 1. Introduction

### 1.1. The Clostridium Neurotoxins

Tetanus is a lethal bacterial infection for all patients without tetanus vaccination. The killing agent is a Zn^2+^-metalloprotease, a protein composed of two amino acid chains called tetanus neurotoxin or tetanus toxin (TeNT). It is produced by *Clostridium tetani* which is related to others *Gram*-positive *Clostridium* like *Cl. botulinum*. Some of these anaerobic bacilli produce the botulinum neurotoxin (BoNT), which is divided into seven serotypes (A, B, C1, D, E, F, and G)—all of which are antigenically distinct, but share the same overall structure with TeTN [1]. In the case of botulinum neurotoxins, humans can be infected by food intake and eventually die. To avoid intoxications, BoNT can be effectively destroyed by cooking food for a few minutes. The thermal treatment is mandatory for industrial food preparations sold in cans. Not all serotypes are life-threatening. Botulism in humans is a disease caused by types A, B, or E, and in rare cases also by type F. In the case of tetanus, the disease takes place through a wound or surgical manipulation without hygienic measures. Both *clostridium* neurotoxins can be lethal to humans and vertebrate animals alike (fish, birds, livestock) in the subnanomolar range acting on the nerve tissue upon internalization (cell uptake) [2].

Both neurotoxins possess specific binding and internalization mechanisms to nerve cells. Since the 80s biochemical assay techniques have evolved, and today, we better understand the pivotal role that human cell surface gangliosides play in internalization [3,4]. The carboxyl-terminal domain of TeNT—AKA “fragment C” or heavy chain (Hc-TeNT)—assists motor nerve cell recognition, binding, and internalization via endocytosis. The process takes place in a highly specific fashion at synapses of cholinergic motor neurons [5,6]. Not only the holotoxin, but also the isolated Hc-TeNT-fragment possesses exceptional molecular mobility in the infected body known as retrograde axonal transport within the motor neuron axons in the direction of the spinal cord [7,8,9]. The finding has the potential to usher in a new area of pharmaceutical research and the development of new drug targeting and delivering systems. The present study reports on the design, synthesis, and bioassays of a novel cyclic peptide to prevent clostridial toxins from cell-surface binding.

### 1.2. The Neurotrophin Receptor Family

Trk stands for Tropomyosin Receptor Kinase and was first identified as an oncogene composed of the 5′ region of the tropomyosin gene fused with a tyrosine kinase domain [10]. Members of the Trk family include TrkA, B, and C; they are highly expressed in cells mainly of neural origins, and they bind neurotrophins (NTs) with different specificity [11]. Like all tyrosine kinase receptors, the Trk receptors are activated by dimerization, resulting in autophosphorylation of intracellular tyrosine residues and the triggering of the signal transduction cascade [12,13].

The TrkA, B, and C neurotrophin receptors possess a significant sequence homology with a conserved domain organization. This conserved structure comprises: The five extracellular domains, a transmembrane region, and the intracellular kinase domain. The extracellular domains consist of a leucine-rich region flanked by two regions rich in cysteine. Domains 4 and 5 are similar to immunoglobulin domains [14]. Studies on TrkB and TrkC have shown that Domain 5 is sufficient for the binding of its ligands and is responsible for their binding specificity [15,16,17,18]. The D5 ectodomain has been widely focused on the elucidation of various crystal structures of human TrkA, B, and C alone or with bound NT and has lent mechanical insights about ligand binding specificities [19]. The intracellular receptor parts embrace a domain with kinase function for signal transduction from the outside, but the limited selectivity of ligands has become an issue for the druggability of the kinase domain, due to highly conserved binding sites among the kinase family members [11].

### 1.3. The Established Receptor Mechanism

The neuronal binding and uptake mechanism of TeNT is still not fully resolved and requires the identification of novel molecular players. Indeed, Lalli et al. [8] demonstrated that the aforementioned carboxyl-terminal domain of tetanus toxin travels travels along motor axons via retrograde axonal transport in neural compartments also comprising, e.g., the p75-NTR. On a molecular level, TeNT interacts as a ligand to nerve-cell membranes via a so-called dual receptor mechanism. It describes the participation of polysialoganglioside receptors (N-acetyl neuraminic acid-containing glycosphingolipids) on the outer leaflet of cell-membrane surfaces in infected organisms [4,20,21,22,23]. The β-trefoil shaped carboxyl-terminal half of the heavy chain (Hcc) provides for cellular receptor and ganglioside binding activities [23,24,25,26,27]. The dual receptor mechanism alludes to the participation of two distinct ganglioside-binding sites on the neurotoxins: (i) The lactose-binding pocket “W”, located around tryptophan W1289 and adjacent residues like histidine H1271 and D1222; (ii) and the sialic acid-binding pocket “R” around arginine R1226 and adjacent residues, like Y1229 [4,23,24,28].

### 1.4. The Inconsistent Dual-Receptor Mechanism

Despite the huge corpus of publications and scientific progress achieved so far, the exact molecular mechanism is still subject to ongoing research and vivid debate [4,9,29,30]. The question has been raised whether gangliosides would constitute the sole binders for the neurotoxins. Taken together with the mainstay of current research literature, it cannot be ruled out that other chemically distinct entities could act as a third binding component to account for the extremely high affinity and specificity, all of which cannot be explained, in principle by the presence of carbohydrate sites alone [9,24,30,31]. It is experimentally known that TeNT, BoNT/A,/B,/C,/E, and/F display affinities in the upper nanomolar range in seminal In Vitro binding assays with immobilized polysialo-gangliosides, albeit clostridial neurotoxins (CNTs) show much stronger affinities to synaptosome preparations of neuronal tissue (one-digit nanomolarity). At a cellular level, removal of sialic acid residues by neuraminidase treatment of cultured cells isolated from the spinal cord [32] or adrenergic chromaffin cells [33] reduced BoNT Type A potency, as well as TeNT action [34]. That and more evidence have been summarized by Rummel and Häfner [31].

Today’s established view describes the high-affinity interaction between Hcc and neuron cells by the aforementioned dual carbohydrate-binding linkage [4,28]. Rummel and colleagues of the Binz group showed that functional “R” (R1226) and “W“ (W1289) binding sites are a prerequisite for TeNT neurotoxicity [23]. This interpretation is in line with the conclusion drawn by Chen and colleagues about the high-affinity binding of TeNT to neurons being mediated solely by gangliosides in the “R” and “W” pockets as dual receptors prior to TeNT entry into motor neurons [4]. The authors mutated the “R” and “W” pockets to demonstrate their role in high-affinity binding. The dual receptor mechanism is implicitly supported by the fact that human neuronal cell surfaces are rich in gangliosides. They show, however, low affinities to the Hc domains of BoNT, and therefore, cannot explain the extraordinary neurotoxicity in subnanomolar ranges in the living organism. Moreover, the binding is sensitive to proteases, strongly suggesting the existence of a protein receptor involvement [34,35]. Further inconsistencies against the dual mechanism are evidenced by the existence of one low and another high-affinity sites in brain tissues and the binding affinity drop from In Vivo to In Vitro bioassays (reviewed in [26]).

### 1.5. The Proposed Receptor Mechanism

In view of the established view of the mechanism [30,36,37], we address the hypothesis that TeNT might mimic NTR-ligands and present preliminary experimental results. In 2020 a seminal work presented a synoptic report concerning newer findings of the mechanistic aspects of cell uptake of tetanus neurotoxins [38]. Recently, two complementary reviews concerning BoNT mechanisms have also been published [39,40].

In need of more compelling explanations, modern literature attests of various efforts to formulate rather complex binding modes incorporating one or two distinct ganglioside contacts with a model of a protein receptor involvement (reviewed in Rummel et al. [31]). In addition, the motor neuron uptake of BoNT/A,/B,/E, and/G was explained by a concert of recognition patterns through ganglioside receptors combined with synaptic vesicle glycoproteins (reviewed in Rummel et al. [31]). Other serotypes interacted in association with synaptic vesicle proteins as an additional receptor component [31]. A tripeptide (Tyr-Glu-Trp) was proposed as a potential ganglioside binding site blocker and found interacting with R1226 at the “R” site under the participation of adjacent residues (D1147, D1214, and N1216) [28]. The advent of a triple receptor mechanism appears eminent, since the relevance of a hitherto unknown protein (receptor?) component for the uptake process has been manifested in the literature for the last two or three decades-although such an additional receptor component has never been identified without ambiguities as outlined in the following [27,34,35].

To a lesser extent, however, the literature in the field of protein modeling has been aware that the primary sequence identity threshold for homologous structure *versus* function predictions must differ. In our case, we have focused on the common Trk binding capabilities (function) of two structurally unrelated ligands, a fact which became evident by the advent of their crystal complexes (e.g., PDB codes: 1HCF [12], 1FV2 [26], 1A8D [27], and 1EPW [41]. Since it has been observed that a common overall structure or domain (fold unit) is more conserved than protein functions—the identity percentage must score even higher than indicated by the established threshold diagram to conserve a specific function in a protein family (more details in the Appendix A). In this regard, postulating a common binding site—the CNTs should have converged into analogous structures of neurotrophins under evolutionary pressure, i.e., in need of recognizing and fitting into the same binding cleft. During eons of vertebral life, not only the natural ligands (neurotrophins), but also the anaerobic clostridial bacteria have optimized their contacts to their host organisms at a molecular scale. This way, CNTs have learned to survive and spread through lethal infection of their hosts, i.e., in oxygen-free soil after cadaver decomposition.

Intriguingly, TeNT shares the same retrograde endocytic route with neurotrophins. They play a physiological role in nerve tissues and are endogenous ligands to their glycosylated neurotrophin receptors (NTRs), which were identified as the tropomyosin receptor kinases A, B, and C (TrkA, B, and C for short) [12]. Based on this finding and J. Aguilera’s experience gathered over decades [9], we assumed that the Trk receptor family is that long-awaited component to account for the aforementioned receptor inconsistencies [9,42,43,44,45,46]. Here we focused on the TrkB receptors to represent all potential neurotoxin target receptors. The neurotrophin ligand binding site (NT-BS) was identified on its fifth extracellular domain (D5 ectodomain) of TrkB [12]. D5 of all three types of Trk receptors are structurally known (cf. PDB codes elsewhere in the text). Their primary sequences are followed by a receptor’s transmembrane segment and the intracellular C-terminal parts [12]. The mechanistic aspects of possible receptor homodimerization are still under debate [47]. It was concluded that the triple receptor mechanism had to be the most plausible solution.

In summary, we consider that we succeeded in demonstrating In Vitro and in silico that a possible receptor for tetanus neurotoxin is the endogenous receptor for Brain-derived neurotrophic factor (BDNF), TrkB; based on this discovery, we have described a synthetic molecule capable of displacing the toxin from its protein receptor, and therefore, it could be a good antidote to tetanus disease.

Altogether, this work combines biochemical and computational methods and explores a possible molecular interaction between TeNT Hc and human TrkB receptors.

## 2. Results

### 2.1. Hc-TeNT Interaction with Target TrkB Receptor

In the three panels of Figure 1, the 3D model of the cyclic peptide is shown together with experimental evidence as a preliminary proof of concept. In Panel (a), the two identical binding segments are highlighted. They should increase the chance of target recognition (more details in SM). Panel (b) shows a dot blot assay with D5 synthetic peptide and Hc-TeNT. The H4 peptide constituted a synthetic peptide with amino acids of D5 peptide in random sequence and was used as a scramble control for the D5 peptide. Hc-TeNT recognized with the anti-His antibody specifically binds to the D5 peptide, but not to the H4 peptide (negative control). Anti-TrkB was used for D5 peptide recognition. In Panel (c), a Western blot showed that Hc-TeNT treatment could induce phosphorylation of the TrkB receptor, although not as strongly as in the case of the treatment with BDNF, and not much different from the nontreated control. Phospho-TrkB concentrations were measured after 10 min treatment with 10 nM Hc-TeNT in cerebellar granule neurons (CGNs). BDNF (50 ng/mL) was used as positive phosphorylation control (label “NT” means nontreated cells). After treatment, cell homogenates were resolved in SDS–PAGE (20 µg of protein per lane), and phosphorylated TrkB in Y705 was detected by Western blot, in parallel with total TrkB and tubulin as a loading control.

In order to biochemically assess the interaction between Hc-TeNT and the TrkB receptor, dot-blot assays were performed to verify the binding of the recombinant Hc-TeNT fragment to a synthetic TrkB-derived peptide. This peptide is based on a cycled repeat of 13 amino acid identical to a region of the D5 domain of TrkB (Figure 1a). The peptide was attached to a nitrocellulose membrane subsequently incubated with an Hc-TeNT solution. Hc-TeNT was recognized with an anti-histidine antibody against the poly-His tag, and the peptide was recognized by an anti-TrkB antibody (Figure 1b). The results show that Hc-TeNT binds to the D5 peptide specifically, since it does not interact with the H4 peptide with a random sequence. A greater amount of peptide attached to the membrane retains higher amounts of Hc-TeNT. Controls without Hc-TeNT or the D5 peptide showed no antibody unspecificity.

We sought to determine if TrkB could act as a functional receptor for Hc-TeNT in CGNs by measuring the receptor phosphorylation after treatment with Hc-TeNT. Cells were incubated in K5 medium for 1 h at 37 °C, and BDNF (50 ng/mL) or Hc (10 nM) were added after 10 min. Treatment of CGNs with 10 nM Hc-TeNT causes moderate phosphorylation in TrkB receptor at 10 min, although it does not reach the phosphorylation level achieved with BDNF treatment (Figure 1c).

As a direct result, Hc-TeNT was able to bind and internalize in CGNs; hence, we analyzed if endocytic vesicles containing Hc-TeNT contained TrkB as well. To this end, CGNs were incubated with Hc-A555 and an antibody against TrkB at 37 °C and allowed to internalize for 1h. After incubation, cells were washed with an acid solution to remove all probes still bound to the neuronal membrane, and TrkB is detected using a fluorescent-labeled secondary antibody (Figure 2a). The intensity profile along the white dotted line in Figure 2a corroborates the overlap between the two proteins in the cytoplasm (Figure 2b). Cell analysis reveals a considerable colocalization between TrkB and Hc-TeNT in vesicles situated along dendrites and soma, with approximately 47% of Hc-TeNT colocalizing with TrkB and 60% of TrkB colocalizing with Hc-TeNT (Figure 2c). This result indicates that both of these proteins were internalized together in the same organelles.

### 2.2. Hc-TeNT Competes with BDNF for Binding and Internalization in Cultured Neurons

To investigate how specific this interaction between Hc-TeNT and TrkB may be, we performed binding assays with Hc-A555 (10 nM) in the presence of different amounts of other competing molecules (Figure 3). As competing molecules, we utilize Brain-derived neurotrophic factor (BDNF), a native ligand for TrkB; Nerve growth factor (NGF), which is a ligand for TrkA, but not for TrkB; and the D5 peptide. Competition experiments show that incubation with higher amounts of BDNF decreases the quantity of membrane-bound Hc, as well as treatment with the D5 peptide. By contrast, NGF does not affect Hc-TeNT binding to cerebellar granule neurons (CGNs) membranes, thus indicating no competition for the receptor. NGF acts as a negative control, since it is a neurotrophin related to BDNF, but does not interact with the TrkB receptor, only with p75^NTR^ and TrkA; this last one not present in this neuronal type [48,49]. Quantification of membrane-bound Hc-TeNT in each condition was performed to obtain the competition curves shown in Figure 3b. Curves were fitted in a nonlinear regression model with GraphPad5, and IC_50_ values for each compound were calculated. BDNF is more efficient in Hc-TeNT displacement (IC_50_ = 3.7 nM) than is the D5 peptide (IC_50_ = 24.2 µM), whereas no competition was observed for NGF. The data strongly supports that Hc-TeNT binds to the TrkB receptor at the neuronal membrane, since a native ligand for TrkB blocks its binding.

We have analyzed whether the uptake of Hc-TeNT is also diminished in the presence of BDNF and D5, since binding was affected. CGNs were incubated with 20 nM Hc-A555 for 1h to allow internalization, and then adding 30 nM BDNF, 30 nM NGF or 20 µM D5. Membrane-bound probes were eliminated with a citrate solution, and immunofluorescence was performed against TrkB (Figure 4a). Results show that the quantity of endocyted Hc-TeNT was not affected by the addition of competitor molecules (Figure 4b). This raises the question of whether the level of colocalization between TrkB and Hc also remains invariable or is Hc-TeNT entering the cells by other mechanisms.

Colocalized pixels are presented in white in Figure 4, while calculated colocalization percentages are shown in Figure 4c. Results indicate that in the presence of BDNF, there is a significant decrease in the percentage of colocalization between Hc-TeNT and TrkB, from 47% to approximately 37%. In parallel, the percentage of TrkB colocalized with Hc-TeNT also decrease, from 61% to 40%. Thus, BDNF would be displacing Hc-TeNT from the TrkB-positive endocytic vesicles. In the NGF condition, the opposite effect is observed; the percentage of overlap between Hc-TeNT and TrkB increases to approximately 57%, so it appears that NGF is facilitating their internalization together. In the presence of the D5 peptide, colocalizing TrkB with Hc-TeNT is significantly reduced, as compared to the control, from 61% to 51%.

### 2.3. Mutations in Predicted Binding Site Affect Binding and Internalization Ability

In order to validate the structural model proposed, three residues were mutated in an Hc-TeNT predicted binding sequence that may have an important role in the interaction with TrkB. Therefore, residues Y266, K311, and E343 were substituted for alanine residues (Y266A K311A E343A) in a triple mutant (Hc-Mut) to break several H-bonds, and saline bridges predicted to exist with TrkB (Appendix A).

The cell association of Hc-TeNT and Hc-Mut was also under scrutiny (Figure 5). To this end, CGNs were treated with Hc-TeNT or Hc-Mut and labeled with AlexaFluor555 (red) for 30 min at 4 °C in binding assays. The cells were fixed in 4% PFA, and cell nuclei were stained with DAPI (blue). The images for panel C of Figure 5 were obtained in the confocal microscope, and the intensities of the red channel were measured under each condition.

Moreover, we analyzed whether Hc-TeNT and TrkB localize together in the NMJ in an In Vivo model simulating the natural infection process. We thus injected the tibialis anterior muscle of adult mice with Hc-A555 and 24 h later immunostained cryosections of this muscle for TrkB and SV2. NMJ were visualized with α-bungarotoxin (α-BTx) that bound to nAchR on the postsynaptic membrane. Results revealed colocalization of Hc-TeNT and TrkB at the presynaptic membrane on NMJ, marked with α-BTx (Figure 6a). Next, it was examined whether these two proteins are internalized and transported together in signaling endosomes. For this purpose, we performed immunofluorescences of the spinal cord lumbar section against TrkB. Twenty-four hours after injection, Hc-A555 internalized at NMJ had already reached the soma of motoneurons innervating the tibialis anterior muscle. Motoneurons somas were detected with Nissl-435/455 staining. Hc-A^555^ and TrkB are located together in a large number of vesicles indicated by white arrowheads in Figure 6b. Control injections with AlexaFluor555 alone show no fluorescence signal. Hence, In Vivo experiments corroborate our hypothesis that Hc-TeNT enters in motoneuron terminals by binding to TrkB and is then retroaxonally transported to the soma together with the receptor.

### 2.4. Computed Results at the Predicted Binding Site

Upon superpositioning of existing target crystal structures with a neurotrophin-liganded ectodomain D5 complex of TrkB, it was deduced that the (binding) hotspot for the hitherto unknown tetanus/TrkB association could constitute the place (spot) on the (clostridial) Hc-TeNT fragment which corresponds to the ectodomain D5 of TrkB (Figure 7). After comparison of the superposed crystal structures of d5/target with d5/NT complexes, it became clear that the assumed protein-protein interface (PPI) of the D5-target effectively resembled that of the known d5/NT complex, despite the fact that CNTs (EC 3.4.24.68 or EC 3.4.24.69) and neurotrophins belong to entirely different fold units. On the target side, the Hc-TeNT fragment weighs 100 kDa approximately, and the carboxyl-terminal half (Hc-TeNT) houses the β-trefoil shaped binding domain (50 kDa) [9]. The neurotrophins in complex with D5 constitute homo or heterodimers of approximately 2 × 25 kDa (PDB code: 1HCF [12]). As members of the cysteine knot superfamily they show three antiparallel pairs of β-strands in connection to four sharp turns (β-hairpins) to form the core structure. The knot is stabilized by three disulfide bonds [50].

Precisely, the complex between the heavy chain C-terminal domain of TeNT and TrkB receptor domain D5 was generated by analogy to the aforementioned complex between neurotrophins NT4/5 and D5 of human TrkB (PDB code: 1HCF [12]). Both ligands belong to absolutely different fold units, but their backbones come close enough only in a few residues. These spatially coinciding residues were first identified upon PPI studies by unsupervised docking and user inspection of surface complementarities. Based on structural analogy, it was assumed that they form a hotspot with a key role for the specific receptor recognition. They create favorable contacts through hydrogen bonding with strong polar and electrostatic contributions and a central hydrophobic groove—the latter should destabilize hydrogen bonding networking of random contacts with other proteins. A binding scheme was drawn to document the detected peculiarity: Neurotrophin and neurotoxin ligand binding analogy by evolutionary convergence in vertebrate organisms.

As a direct result of analogy modeling, binding-relevant residues were identified on both sides of the protein-protein interface. On the ligand side, TeNT residues K1174, Y1202, N1203, and N1204 could be identified as analogous to R144, L33 and R34 of NT. They belong to a larger contact zone: E1127, Y1129, K1174, S1201, Y1202, N1203, N1204, E1206, E1310, and D1315.

On the NTR receptor side, the primary sequence segments of different Trk types, as well as vertebrate species, were compared to assess the expected high degree of conservation among the NT and CNT-binding residues (Appendix A).

Before proposing residues for mutational studies as a proof of concept, the PPI model was analyzed, and critical aspects summarized (Appendix A). Of note, the experimental data, including the mutational studies, were documented elsewhere (for more details, refer to patent application “Peptide for use in the treatment of disease caused by clostridium neurotoxins”, WO 2017050816 A1).

The analogy modeling approach reflects that neurotrophins and CNTs are not related by homology. However, they share binding similarities by analogy. Both ligands recognize and bind to domain D5 of the Trk family (all-beta or Ig-like fold). Upon D5 binding, their geometries coincide only in a few surface residues in proximity to the proposed PPI hotspot (Appendix A).

In the next step, the primary structure of the proposed PPI model was searched by BLAST against the UniProt data bank of nonredundant protein sequences to study the homology of the Trk family in the same species or across different animals. To this end, a 14-residue long segment of D5 was under scrutiny for conserved positions: GCLQLDNPTHMNNG (cf. Graphical abstract). The aligned segments of vertebrate animals showed that the surrounding local region is highly conserved. The 14-residue long segment is identical (Appendix A). Its structure is preserved, since its function has not changed despite the divergent development of those vertebrate species. The three neurotrophin receptor subtypes—TrkA, B, and C—are also highly conserved in this part.

The NTR segments are highly conserved, although not totally identical. Hence, to enhance the theoretical antidote effectiveness, only a minor need arose to design three individuals for TrkA, B, and C. As sequence alignment studies showed there was no need for species-specific antidotes because the GCLQLDNPTHMNNG segment remains unchanged for Mammalian animals and chicken (Appendix A).

Prior to the experimental evaluation, the synthesis work was carried out by H.-R. Rackwitz at Peptide Specialties Laboratory, Heidelberg, Germany, who also produced a scrambled version as a negative control for the biochemical assays (cf. Section 4.2 “Synthesis of the cyclic peptide and negative control sequences”).

The main implication of the aforementioned finding for the experimental work was that TrkB would constitute the target under scrutiny in the present study concerning the binding of endogenous neurotrophins, as well as clostridial neurotoxins alike. As mentioned above, both ligand types are totally unrelated in sequence composition and structure. The former bind as dimers, while the latter as monomers with their Hc-TeNT parts. However, following our assumption, they could share the same binding site on the Trk neurotrophin receptors. In more general terms, to date, it has remained a computational challenge to determine which amino acids is given positions of aligned sequences with low homology are signaling or just contributing to the noise. Sequence variations of biomolecules between and within species are either relevant (signal) or irrelevant (noise) to explain that variability in structures and functions. Noise comes from either random mutation without relation to structure and function during evolution or from experimental or data handling errors. In our study, interacting residues were postulated by design for wild type and mutant type experiments. After clarifying the analogous binding to ectodomain D5, which could occur prior to internalization (cell uptake)–the small cyclic D5 peptide was designed as a novel neurotoxin inhibitor and tested as a starting point for a potential antidote development (Appendix A).

In Figure 8, it can be observed by the naked eye that even if the main chain fragments (near the black C-Term label CT) of arginine (beige R) and lysine (blue K) appear to be in proximity (at least locally in this binding patch), they soon follow completely different directions, since the overall folds of both proteins are totally different. To the right and topmost, it can be appreciated how two asparagine residues (blue N+N) replace the intensive bonding network the arginine (beige R) provides. At that site, both backbones (blue and beige) fairly differ because both asparagine residues only come into the interaction spot by their (omega) head groups, the positions of their C alpha atoms on the backbones absolutely do not coincide with the orientation and position of the analogous arginine. The hydrophobic zone of the binding patch is colored green (Figure 8).

## 3. Discussion

Despite a large number of publications describing the infective process and possible diverse receptors for tetanus toxin, there is no evidence yet about the exact mechanism utilized for the highly specific entry of the toxin. Several protein receptors have been proposed, such as the membrane glycoprotein Thy-1 or the synaptic vesicle protein SV2; however, the involvement of these proteins in the endocytosis of the toxin in the NMJ has not been demonstrated [22,51]. In this work, preliminary evidence hints at the interaction between TeNT and neurotrophin receptors Trk, by in silico and biochemical approaches. Molecular modeling reveals the high analogy between neurotrophins and TeNT in the binding to Trk receptors, and the potent noncovalent interactions that could be established between TeNT and Trks, especially TrkB. In this model, W and R pockets in the Hc-TeNT structure are completed with a novel binding site located in a loop in the Hcc subdomain. This new binding site fits perfectly to the extracellular D5 domain in TrkB, which also acts as a binding site for neurotrophins [52]. Therefore, TeNT could interact at the same time with two PSG molecules and the TrkB receptor, since this binding site is far enough from the other two pockets, in line with the dual receptor theory regarding high affinity binding [20]. The interaction between TeNT and the D5 domain of TrkB is corroborated by the In Vitro binding between the D5 peptide, based on the D5 domain sequence, and the Hc-TeNT fragment. Results also confirm the important role of Hc-TeNT residues Y266, K311, and E343 in this interaction, since there is a loss in binding and uptake capacity in the mutated fragment. Nevertheless, other residues may be involved because the interaction is not completely abolished, and the fragment is still capable of internalizing.

Moreover, the interaction between Hc-TeNT and TrkB is functional, since it has an effect on the activation of the receptor. In previous work, the group had observed that both the whole TeNT and the Hc-TeNT fragment were able to phosphorylate Trk receptors and activate downstream signaling in cortical neurons and cerebellar granule neurons in culture and in rat synaptosomes [42,43,44,45,53]. A similar mechanism has been described for BoNT/A, which can bind the FGFR3 receptor and promote its phosphorylation in the same manner as FGF2, its natural agonist [54]. Furthermore, FGFR3 ligands can block BoNT/A binding and internalization in neuronal cells. This finding strongly supports our hypothesis, since FGFR3 belongs to the tyrosine kinase receptors family, as well as Trk receptors, and BoNT/A is highly homologous with TeNT.

In our experiments, BDNF, but not NGF, can compete with Hc-TeNT in the binding to neuronal membranes, indicating competition for the same binding site. These results are contrasted to those obtained by Roux and collaborators describing that muscular coinjection of Hc-TeNT and BDNF in mice facilitates Hc-TeNT internalization in NMJ [55]. The same effect was observed for NT-4/5, but not as pronounced. These differences could be due to the different BDNF concentrations assayed, since the enhancement in Hc-TeNT internalization was observed in a range of 2.5 ng–100 ng of BDNF, while we work at higher concentrations. It has also been reported that low doses of BDNF increase synaptic transmission in NMJ in depolarizing conditions, which could mask a possible competition effect of BDNF by enhancing Hc-TeNT uptake through activity-dependent pathways [56,57].

We also have observed that an elevated concentration of the D5 peptide can displace membrane-bound Hc-TeNT, presumably by blocking its Trk-binding site. This is an interesting result, since the peptide could be used as a blocking agent for TeNT in tetanus infected patients, instead of the current vaccine. It will have important advantages, as compared to the tetanus toxoid vaccination, for example, a more economical and secure production, as well as fewer secondary effects, produced mostly by the formaldehyde used in the toxin inactivation process.

It is known that Hc-TeNT possesses retrograde transportation utilizing the same endocytic pathway as neurotrophins and their receptors. In cultured motor neurons, Hc-TeNT shares signaling endosomes with p75NTR and TrkB, as well as NGF and BDNF, and the conversion of Rab5 to Rab7 GTPase is a key step in the endosome maturation and its intracellular sorting [58,59,60]. In this work, we corroborate these results in CGNs and in an In Vivo system. In the CGNs cellular model, approximately 50% of Hc-TeNT positive endocytic vesicles contain TrkB. The remaining Hc-TeNT could be entering neurons by using another protein receptor or by an activity-dependent synaptic vesicle recycling mechanism, as described for cortical and hippocampal neurons [51,61]. CGNs require high K+ concentration in the medium to survive, resulting in depolarizing membrane conditions that promote synaptic vesicle recycling. In the same way, after an intramuscular injection in mice, Hc-TeNT colocalizes with the TrkB receptor (p75NTR and Rab5) at the presynaptic side in the NMJ. These observations are opposite to the results published by Roux in which there is no colocalization observed between TrkB and Hc-TeNT in NMJ 30 min after an intramuscular injection in the LAL muscle [55]. At 30 min postinjection, Hc-TeNT remains in NMJ, and it has not begun the ascent along the axon yet, whilst 24h postinjection most of the Hc-TeNT has been transported to the motor neuron soma. Then it is possible that this difference in the experimental time could explain the discrepancies in these results. We also observed elevated coincidence of TrkB and Hc-TeNT, and between Hc-TeNT and Rab5, at the soma of motor neurons innervating the tibialis muscle, indicating that all of them share endocytic compartments during axonal retrograde transport In Vivo.

Our work was based upon experimental findings concerning retroaxonal transport of Hc-TeNT. Neurotrophin receptors have been identified along with various proteins, such as Vps26, SNX1, and BICD1, which have been described to be involved in their intracellular trafficking and sorting [60,62]. Although experiments by Terenzio et al. were carried out in the absence of exogenous neurotrophins, Trk receptors and p75NTR were found in signaling endosomes. Under nonactivation conditions, neurotrophin receptors usually are endocytosed by a clathrin-independent local pathway [63]. This result agrees with our hypothesis as the Hc-TeNT present in the experiment could be activating the receptors and directing them into a retroaxonal transport pathway.

A recent study identifies nidogen, a basal lamina protein, as a receptor for TeNT at the NMJ [64]. Nidogen could be acting as a concentrator of TeNT in certain regions of the plasmatic membrane where PSG and other transmembrane proteins would permit the binding and internalization of the toxin. This mechanism is shared with different growth factors, including neurotrophins, and this binding to the extracellular matrix proteins is a critical step for its biological function and signaling [65,66,67]. These results may be compatible with the model proposed in this paper as the nidogen cannot directly mediate endocytosis and would need a membrane receptor that triggered this process. The authors suggest that one of these receptors could be tyrosine phosphatase LAR (Leucocyte Common Antigen-related receptor). It has been observed that LAR can interact with nidogen, as well as with Trk receptors, regulating their downstream signaling [68,69,70,71]. Thus, a connection between nidogen and TrkB could be established. Nidogen would bind to TeNT by its R pocket and concentrate the toxin at presynaptic lipid rafts in NMJ. Here TeNT would interact with PSG through its W pocket and to the D5 domain of TrkB by the novel binding site proposed in this paper. This hypothesis also agrees with Chen and collaborators’ results, defending that TeNT high affinity binding requires that the two ganglioside binding pockets be occupied [4].

In the model we propose, TrkB would act as a receptor for TeNT at axonal terminals of motor neurons. However, TrkB would not be the only TeNT receptor, since it has been observed that Hc-TeNT can be internalized into PC12 cells, which do not express TrkB, activating specific signaling pathways. Additional evidence exists that Hc-TeNT can promote nSMase activity through the p75NTR receptor [9]. Therefore, it is possible that, as neurotrophins, TeNT could bind more than one neurotrophin receptor and present different affinities for each one. In fact, molecular modeling analysis indicates potential interactions with the D5 domain of TrkA and TrkC, and TrkA activation in response to TeNT has been described previously by our group [42]. Future experiments should be performed to determine the possible implication of the others neurotrophin receptors in the infective process of TeNT.

## 4. Materials and Methods

### 4.1. The Peptide Design to Evaluate the Protein-Protein Interface

The design of an inhibitor was based upon the ectodomain D5 as a potential receptor. It belongs to the cell adhesion class of proteins (all-beta fold). Thus, it has evolved during eons of evolutionary times with conserved peptide or protein recognition sites on its surface. Three seminal reports succinctly reviewed the major tenets about PPI inhibitor development [72,73,74]. The other design rationale was to build a cyclic molecule; thereby, enhancing metabolic stability, while conserving most of the (predicted) C-alpha orientations of the backbone in addition to the (predicted binding) active conformations of the side chains (Scwrl 4 rotamer library). The interacting segment of D5 is a linear peptide with a concave curvature to form a hemicycle with three hydrophobic side chains turned inward. By duplicating that peptide string and rotating one half about 180° around the ring center, both halves could be joined without distortion of the backbones. In addition, the ring formation should reduce most of the unproductive conformational (degrees of) freedom and “freeze-in” the active conformation(s) from the native structure. The chemical cyclization was devised by a Cys-Cys (disulphuric) bridge after mutating the positions of Cys and Met in the original peptide segment under Swiss PDB Viewer [75].

The residue composition was compared to a small group of residues which are almost always found at protein-protein interfaces [76]. Two (P, D) out of six (P, I, Y, W, D, and R) participate in the present PPI. Another report empirically collected the most conserved residues in binding pockets (not necessarily shallow surface depressions). From those six residues (G, E, R, D, H, and T), three are also present in the designed peptide (D, H, and T) [77]. The binding mode comprises an elaborated hydrogen bond network, a typical feature prevailing in contact zones between physiological compounds and in stark contrast to drugs-like ligands (nonpeptides) [77].

### 4.2. Synthesis of the Cyclic Peptide and Negative Control Sequences

The peptide was synthesized on a continuous flow synthesizer using Fmoc Solid Phase Peptide Chemistry [78]. The peptide amides were synthesized on Rink Amide AM resin (200–400 mesh, 0.62 meq/g, Merck-Millipore, Burlington, MA, USA) and peptide acids on Tritylchloride Polystyrene(TCP)-resin loaded with the C-terminal Fmoc-L-amino acid (200–400 mesh, 0.5 meq/g, Intavis AG, Cologne, Germany). The sequences were assembled in a stepwise fashion from the C- to N-terminus using Fmoc protected L-amino acids with trityl or tert-butyl side chain protection, PyBOP as a condensation reagent, N-methylmorpholine as base, and DMF as solvent. The Fmoc protecting group was removed with 25% piperidine in DMF. Upon completion of the chain assembly peptides, were cleaved off the resin with 95% TFA, 4% triethylsilane, 1% water. Crude products were dissolved in 15% acetonitrile in 0.1% aq TFA and purified by reversed phase HPLC on a C18 column (Phenomenex, Torrance, CA, USA) using a linear gradient from 10% to 80% acetonitrile in 0.1% aq TFA. The resulting purified fractions were analyzed by MALDI TOF mass spectrometry and lyophilized yielding the corresponding TFA salts.

### 4.3. Expression and Purification of Hc-TeNT

*Escherichia coli* BL21 cells containing the expression vector pYN1, a pQE3-derived expression vector (Qiagen; Chatsworth, CA, USA), containing the cDNA for the Hc-His_6_ fusion protein (Hc-TeNT), were grown in Luria-Bertani medium containing 100 µg/mL ampicillin. Protein expression was induced by the addition of 0.4 mM isopropyl β-D-thiogalactoside (IPTG). Hc-TeNT purification was performed as reported previously [53].

In some experiments Hc-TeNT was labeled with AlexaFluor (Thermo Fisher Scientific, Waltham, MA, USA) following the manufacturer’s instructions obtaining Hc-TeNT AlexaFluor^555^ (Hc-A^555^).

### 4.4. Cloning and Expression of Hc-Mut with a Triple Mutation (Y266A-K311A-E343A)

A construct of 905 pb consisting of a C-terminal segment of Hc-TeNT was synthesized to replace for alanines positions Y266, K311, and E343 (amino acid number in Hc-TeNT domain sequence). This fragment was cloned in the pMK vector (Thermo Fisher Scientific, Waltham, MA, USA) using SacII and KpnI restriction sites, thus obtaining the plasmid pMK-Hc (3205 pb, kanamycin resistance). This construct contains bases 431 to 1221 of the Hc sequence, including several interspersed repeats of the same sequence that allow for the inclusion/exclusion of the selected amino-acidic positions in both wild type and mutated forms. To obtain the triple mutant Hc-Y266A-K311A-E343A (referred to as Hc-Mut), pMK-Hc was digested with SalI and then ligated to remove the fragment containing K311, and then the plasmid was digested with MluI and ligated to remove E343. Finally, the resulting plasmid was digested with SacII and KpnI, and the digestion product was cloned in the pYN1 vector to yield the pYN1-Mut expression vector. The introduction of the correct mutations was evaluated by DNA sequencing of recombinant clones. The recombinant plasmid was transformed into *Escherichia coli* BL21 (Qiagen; Chatsworth, CA, USA), and the expression of the mutated protein was performed in the same way as described in the previous section.

### 4.5. Cerebellar Granule Neurons (CGNs) Cultures

Primary cultures of CGNs were prepared from 7-day-old Sprague-Dawley rats obtained from the *Servei d’Estabulari of the Universitat Autònoma de Barcelona* (Barcelona, Spain). Cerebellar tissue was dissected free of meninges, chopped into little pieces, and incubated for 10 min at 37 °C in a Krebs-Ringer buffer solution with 0.025% trypsin (Sigma-Aldrich, St. Louis, MO, USA) and 0.3% bovine serum albumin (BSA, Sigma-Aldrich, St. Louis, MO, USA). The reaction was stopped by the addition of soybean trypsin inhibitor (0.6 mg/mL, Thermo Fisher Scientific, Waltham, MA, USA) and deoxyribonuclease I (0.08 mg/mL, Sigma-Aldrich), followed by centrifugation. Cells were dissociated by repeated pipetting and separated from nondissociated tissue by a 100-μm nylon mesh. The cell suspension was seeded at a density of 7 × 10^4^ cells/cm^2^ in 24-well plates previously coated with poly-L-lysine (10 µg/mL). Cells were grown in BME medium (Thermo Fisher Scientific, Waltham, MA, USA) supplemented with 10% fetal bovine serum, 25 mM KCl, 5 mM glucose, 2 mM glutamine, penicillin (50 UI/mL), and streptomycin (50 μg/mL), referred to in the text as K25 medium. Cytosine arabinoside (10 μM) was added 24 h after plating to prevent the proliferation of nonneuronal cells, mainly astrocytes and microglia. The cultures were maintained for seven days In Vitro (DIV), when the CGNs were considered fully differentiated, and assays were performed. For some experiments, cells were transferred to a 5 mM KCl serum-free BME medium (referred to as K5 medium) supplemented with 5 mM glucose, 2 mM glutamine, penicillin (50 UI/mL), and streptomycin (50 μg/mL).

### 4.6. Western Blot Analysis

After treatments, cells were recollected in lysis buffer (50 mM Tris-HCl pH 6.8, 2mM EDTA, 0.5% Triton X-100) supplemented with a cocktail of protease and phosphatase inhibitors (Roche). Samples were sonicated for 15 s in ice, and their protein concentration was determined by the bicinchoninic acid assay (BCA, Thermo Fisher Scientific, Waltham, MA, USA). Samples mixed with loading buffer (70 mM Tris-HCl pH 6.8, 1% SDS, 11% *v*/*v* glycerol, 0.01% β-mercaptoethanol, 0.25% bromophenol blue) were loaded in 10% polyacrylamide gels, and sodium dodecyl sulfate polyacrylamide gel electrophoresis (SDS-PAGE) was performed (50 µg of protein per lane). After the electrophoresis, proteins were transferred to nitrocellulose membranes at 100 V for 45 min in a transfer buffer that contained 25 mM Tris-HCl, 0.2 mM glycine, 10% (*v*/*v*) methanol. Membranes were blocked with 6% nonfat dry-milk solution in TBT-T (140 mM NaCl, 2.7 mM KCl, 4.3 mM Na_2_HPO_4_·H_2_O, 1.5 mM KH_2_PO_4_, 0.1% Tween 20), and afterward, membranes were incubated with the corresponding primary antibody diluted in a BSA 5% solution; using the following: Anti-rabbit *phospho*-Y705 TrkB (1:1000; ab52191; Abcam, Cambridge, UK), rabbit anti-TrkB (1:1000; ab33655; Abcam, Cambridge, UK) and mouse anti-β-tubulin (1:10,000; 556321; BD Biosciences). Membranes were incubated for 1h with a secondary antibody conjugated with horseradish peroxidase diluted to 1:2000. Western blots were developed using a 1:1 chemoluminiscent mixture [1M Tris-HCl pH 8.5, 0.5 M luminol, 79.2 mM p-coumaric acid: 1M Tris-HCl pH 8.5, 8.8 M H_2_O_2_], and exposed to ECL film membranes (Amersham Pharmacia Biotech; Buckinghamshire, UK). Analysis of the resulting bands was performed using Quantity One (Bio-Rad Laboratories, Hercules, CA, USA).

### 4.7. Dot-Blot Analysis

The D5 peptide (10 µg and 20 µg) and H4 peptide (10 µg and 20 µg) were attached to a nitrocellulose membrane by applying it drop by drop. The membrane was blocked with 5% nonfat dry-milk solution in TBT-T for 1h and then washed with TBT-T. Membranes were incubated with 1 mg/mL Hc-TeNT for 1h, and afterward, primary antibodies diluted in a BSA 5% solution were added o/n at 4 °C. Antibodies used were mouse anti-polyhystidine (1:1000; 27-471001; Amersham Pharmacia Biotech, Buckinghamshire, UK), rabbit anti-TrkB (1:1000; ab33655; Abcam, Cambridge, UK) and mouse anti-β-tubulin (1:10,000; 556321; BD Biosciences). Secondary antibodies used were anti-rabbit IgG HRP conjugate (Thermo Scientific), and anti-mouse IgG HRP conjugate (Bio-Rad Laboratories, Hercules, CA, USA) diluted 1:3000 in TBS-T plus 5% nonfat dry-milk. Membranes were developed using a 1:1 chemoluminiscent mixture [1 M Tris-HCl pH 8.5, 0.5 M luminol, 79.2 mM p-coumaricacid: 1 M Tris-HCl pH 8.5, 8.8 M H_2_O_2_], and exposed to ECL film membranes (Amersham Pharmacia Biotech Buckinghamshire, UK). Analysis of the resulting dots was performed using Quantity One (Bio-Rad Laboratories, Hercules, CA, USA).

### 4.8. In Vivo Hc-TeNT Intramuscular Injections

All animal experimentation procedures were performed according to the Committee for Animal Care and Use of the University of Lleida, in line with the Generalitat de Catalunya Norms (DOGC 2073, 1995). Two months old CD1 mice purchased from Harlan Laboratories (Castellar del Vallès, Catalonia, Spain) were anesthetized with ketamin/xylacin, and afterward, intramusculary microinjected in the tibialis anterior muscle with Hc-A555 (5 µg, 7 µL total volume) or an equivalent volume of AlexaFluor555 alone, as a negative control. After 24h, animals were anesthetized and transcardially perfused with saline followed by 4% paraformaldehyde (PFA) in 0.1 M (pH 7.4) phosphate buffer. The tibialis anterior muscle and spinal cord lumbar section were dissected, fixed in 4% PFA at 4 °C (for 30 min for muscle tissue and 24 h for the spinal cord tissue), and cryoprotected with 30% sucrose in 0.1 M phosphate buffer and 0.02% sodium azide (pH 7.4). Tissues were mounted in O.C.T. (Tissue-Tek) and frozen at −80 °C before being sectioned on a cryostat (30 μm thickness for muscle tissue and 16 μm for the spinal cord) and collected onto poly-lysine coated slides.

### 4.9. Immunohystochemistry and Image Analysis

Muscle and spinal cord sections were permeabilized with PBS (137 mM NaCl, 2.7 mM KCl, 10 mM Na_2_HPO_4_, 1.8 mM KH_2_PO_4_; pH 7.4) containing 0.1% Triton X-100 for 1 h, then blocked with 10% normal horse serum (NHS; Sigma-Aldrich, St. Louis, MO, USA) in PBS for 1 h, and incubated o/n at 4 °C with primary antibodies diluted in blocking solution: Rabbit anti-TrkB (1:1000; ab33655; Abcam, Cambridge, UK) or mouse anti-SV2 (1:1000; Kathleen M Buckley, Harvard Medical School). After being washed with PBS 3 times, sections were incubated at room temperature for 1 h with the appropriate secondary antibody; either AlexaFluor488 anti-rabbit was used (1:500; Thermo Fisher Scientific, Waltham, MA, USA) or DyLight 405 anti-mouse (1:100; Thermo Fisher Scientific). In muscle preparations, α-bungarotoxin coupled to AlexFluor488 (1:500; Thermo Fisher Scientific Waltham, MA, USA) was incubated together with secondary antibodies. Spinal cord sections were labeled with fluorescent NeuroTrace 435/455 Nissl staining (1:150; Thermo Fisher Scientific Waltham, MA, USA). Finally, preparations were mounted with a laboratory-made antifading mounting medium containing 0.1 M TrisHCl buffer (pH 8.5), 20% glycerol, 10% Moviol, and 0.1% 1,4-diazabicyclo [2,2,2]octane. Digital images were obtained using Fluoview FV10i (Olympus, Hamburg, Germany) or Zeiss LSM 700 (Zeiss, Germany) confocal microscopes, at 0.5–1 µm-thick Z step. Image analysis was performed using ImageJ software (U.S. National Institutes of Health, Bethesda, MD, USA) or Bitplane (Imaris, Bitplane, CT, USA).

### 4.10. Hc-TeNT Endocytic Assay and Uptake Competition Assays in CGNs

CGNs seeded upon coverslips were treated with 20 nM Hc-A555 for 1h at 37 °C, and an anti-TrkB primary antibody (1:500; ab33566; Abcam, Cambridge, UK). In competition assays, the D5 peptide (20 µM, molar ratio 1:1000), BDNF (400 ng/mL, 15 nM) and NGF (400 ng/mL, 15 nM) were added. In the case of peptide D5, it was preincubated with Hc-A555 at room temperature for 1h and then simultaneously added into the culture. Cells were washed with ice-cold PBS followed by an acidic solution (10 mM Citrate-NaOH, 140 mM NaCl, pH 2) for 1 min to eliminate membrane-bound Hc-TeNT fluorescence. Cells were fixed in 4% PFA for 20 min and permeabilized with 0.1% Triton X-100 in PBS for 10 min. After blocking with 5% BSA for 1h, secondary antibody diluted in 5% BSA was incubated; AlexaFluor488 anti-rabbit (1:1000; Thermo Fisher Scientific Waltham, MA, USA). Nuclei were stained with DAPI (0.5 µg/mL; Thermo Fisher Scientific, Waltham, MA, USA) for 5 min, and finally, preparations were mounted in FluoromountG (Southern Biotech¸ Birmingham, AL, USA). Computer images were obtained using a Zeiss LSM 700 (Zeiss, Germany) confocal microscope. Colocalization analyses were performed using Imaris software (Imaris, Bitplane, CT, USA), and fluorescence intensities were measured with ImageJ software (U.S. National Institutes of Health, Bethesda, MD, USA).

### 4.11. Binding Competition Assays

CGN were seeded upon coverslips and 7 DIV later were treated with 10 nM Hc-A555 for 30 min at 4 °C together with increasing amounts of BDNF (50, 200 and 400 ng/mL), NGF (50, 100, 200, 400 and 800 ng/mL) and D5 peptide (5, 25, 50, 200, 400 and 600 μg/mL). After incubation time cells were fixed in 4% PFA and nuclei were stained with DAPI (0.5 µg/mL; Thermo Fisher Scientific, Waltham, MA, USA) for 5 min. Finally, preparations were mounted in FluoromountG (Southern Biotech, Birmingham, AL, USA) and digital images were obtained using a Zeiss LSM 700 (Zeiss, Germany) confocal microscope. Integrated fluorescence intensity (Fluorescent Area × Fluorescence Intensity) was determined using ImageJ software (Imaris, Bitplane, CT, USA). Data were processed with GraphPad Prism 5 (GraphPad Software, La Jolla, CA, USA) to obtain competition curves and to calculate inhibition constants (IC_50_) using this equation: Y = (Y^f^ + (Y^0^ − Y^f^))/(1 + 10^X−LogIC50^). Y, intensity; X, competitor concentration; Y^f^, intensity at maximum competitor concentration; Y^0^, intensity without competitor.

### 4.12. Flow Cytometry in Hc-TeNT Binding and Uptake Assays in CGNs

CGNs were seeded in 24-well plates for 7 DIV and then incubated with 500 nM Hc-A555 or 500 nM Hc-Mut-A555, for 30 min at 4 °C in binding assays and for 2 h at 37 °C in uptake assays. Cells were analyzed with a flow cytometer (Cytomics FC500, Beckman Coulter; CA, USA), 30,000 events were counted in each condition, and their fluorescence intensity was measured at 565 nm in an FL2 detector. The resulting data were processed with CXP software (Beckman Coulter; CA, USA) to obtain histograms showing the number of events in each value of fluorescence intensity.

### 4.13. CGNs Immunocytofluorescence

Immunofluorescence was performed in CGNs for comparison studies between Hc-TeNT and Hc-Mut colocalizing with TrkB. CGNs were seeded upon coverslips and grown for 7 DIV and incubated with Hc-A555 or Hc-Mut-A555 (10 nM each one), for 30 min at 4 °C in binding assays and for 2h at 37 °C for uptake assays. After washing with ice-cold PBS, cells were fixed in 4% PFA for 20 min and incubated with blocking solution (10% horse serum, 0.2% Triton X-100, 10 mM glycine in PBS) for 1 h. Primary antibody anti-TrkB (1:1000; ab33566; Abcam, Cambridge, UK) diluted in blocking solution was incubated o/n. After washing with PBS + 0.2% Tween-20 secondary antibody was added after 1h; AlexaFluor488 anti-rabbit kit was used (1:1000 in blocking solution; Thermo Fisher Scientific, Waltham, MA, USA). Nuclei were stained with DAPI (0.5 µg/mL; Thermo Fisher Scientific, Waltham, MA, USA) for 5 min, and finally, preparations were mounted in FluoromountG (Southern Biotech¸ Birmingham, AL, USA). Digital images were obtained in a Zeiss LSM 700 (Zeiss) confocal microscope. Colocalization analyses were performed using ImageJ software (U.S. National Institutes of Health, Bethesda, MD, USA).

### 4.14. Molecular Modeling and Bioinformatics

The following programs were used for our in silico study: SPDBV [75], Vega ZZ [79], Chimera alpha 1.14 [80], AutoDock 4.2 combined with MGL Tools 1.5.6 [81], as well as PyMol 0.99 [82].

## 5. Conclusions

Exploring computationally and experimentally possible interaction of TrkB in an early stage of the cell uptake mechanism, which leads to subsequent retrograde axonal transport, is a way to contribute to the ongoing discussion about how pathogens and viruses can invade the human body and how remedies may be conceived against those clostridial neurotoxins. For instance, the rabies virus (RABV) and cholera toxin (CTx) are also transported retroaxonally like TeNT, and canine adenovirus t/Type 2 (CAV-2) share the same endocytic carriers with TeNT. The identification of the receptor, or receptors, for TeNT is a key aspect for medical applications and toxicology, in addition to providing new strategies against tetanus. It will have an important impact on the developing of new therapies against neurodegenerative diseases, like Parkinson’s or ALS, by the utilization of Hc-TeNT as a vector for the delivery of therapeutic molecules to the CNS, or by using Hc-TeNT itself as a drug, taking advantage of its observed therapeutic properties.

## Figures and Tables

**Figure 1 molecules-26-03988-f001:**
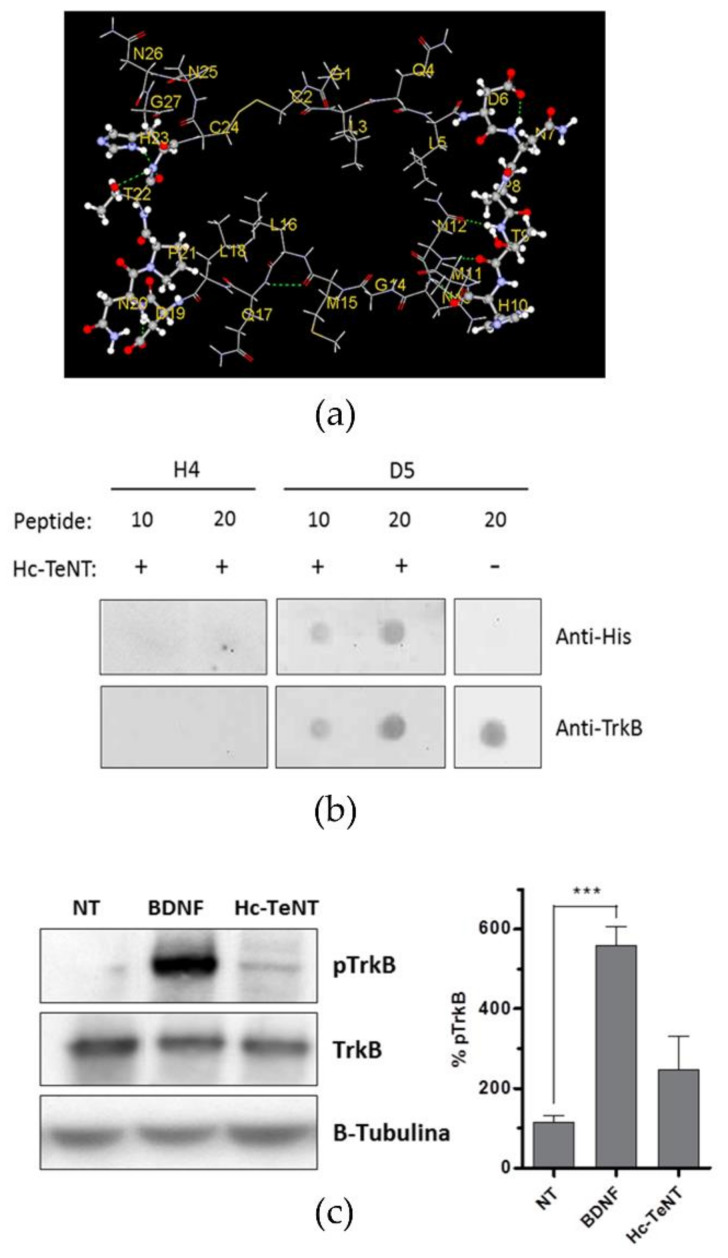
Hc-TeNT interacts with the TrkB receptor. (**a**) 3D-model of synthetic D5 peptide. (**b**) Dot blot assay with D5 synthetic peptide and Hc-TeNT. H4 peptide was used as a negative control for the D5 peptide. Hc-TeNT recognized with the anti-His antibody specifically binds to the D5 peptide and not to the H4 peptide. The anti-TrkB complex was used for D5 peptide recognition (**c**) Bar plot of phospho-TrkB levels after treatment with Hc-TeNT in cerebellar granule neurons. The bar plot (right) shows the quantification of pTrkB signal standardized by the amount of β-tubulin (loading control), in a total of 5 independent experiments. *** *p* < 0.001 using one-way ANOVA followed by Dunnett multiple comparison test.

**Figure 2 molecules-26-03988-f002:**
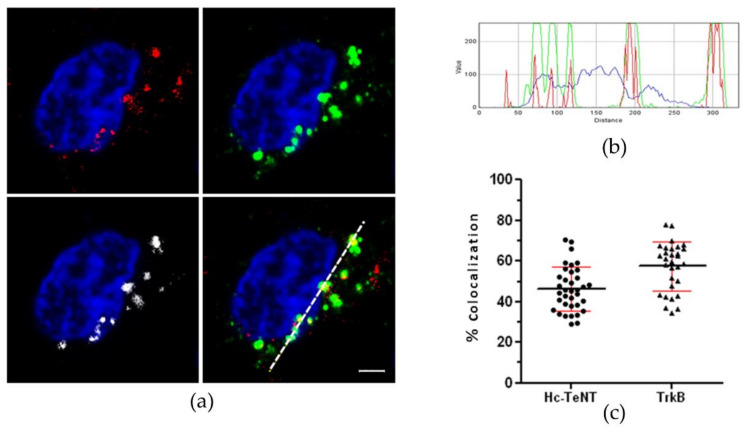
Colocalization between endocyted Hc-TeNT and TrkB in CGNs. (**a**) Immunofluorescence of cerebellar granule-neurons (CGNs) treated with 20 nM Hc-A555 for 1h at 37 °C. TrkB was deTable 488. secondary antibody, and nuclei were stained with DAPI. Colocalizing points were shown in white (left bottom panel). Scale bar = 2 μm. (**b**) Fluorescence intensity profile for the different channels along the white dotted line in (**c**). Colocalization percentages for Hc-TeNT and TrkB calculated in Imaris. Thirty-five fields of two independent experiments were analyzed.

**Figure 3 molecules-26-03988-f003:**
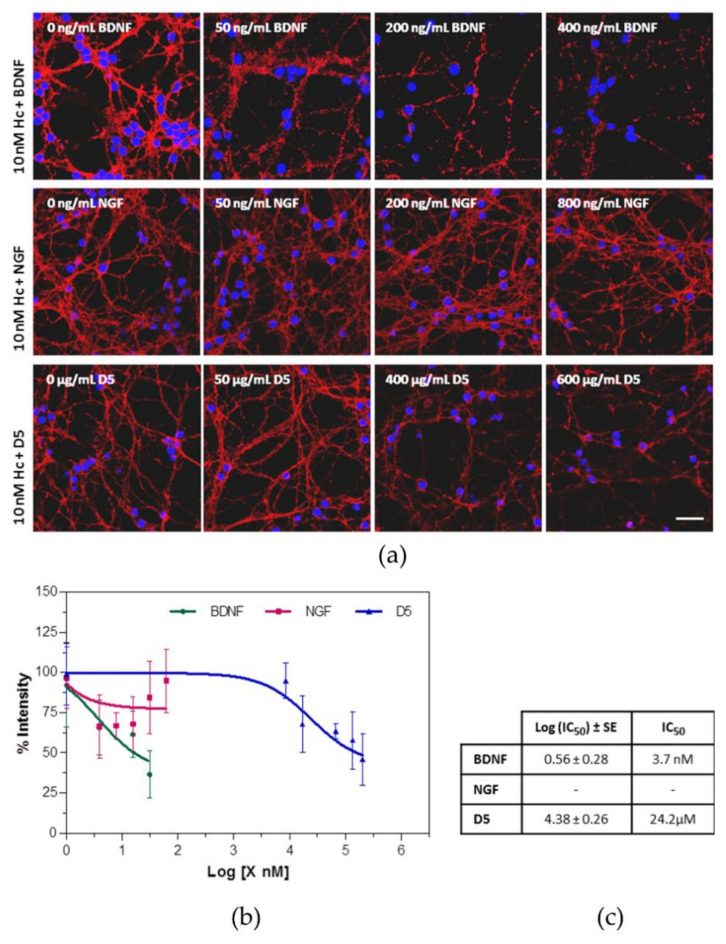
BDNF and the D5 peptide compete with Hc-TeNT for membrane binding. (**a**) Representative confocal images of CGNs. nM Hc-A555 (red) and increasing concentrations of BDNF, NGF, and D5 peptide for 30 min at 4 °C. Cells were fixed, and nuclei were stained with DAPI (blue). Increasing concentrations of BDNF and D5 peptide, but not NGF, reduced the amount of Hc-TeNT bound to the cell surface. (Scale bar = 20 μm). (**b**) Hc-A555 intensity was calculated in each condition, and competition binding curves were obtained. The mean and standard deviation of each condition were shown in the graph. Color code: Competition with BDNF in green, with NGF in red, and in blue for D5 peptide. (**c**) Half maximal inhibition concentrations (IC_50_) for each curve were calculated with GraphPad Prism software.

**Figure 4 molecules-26-03988-f004:**
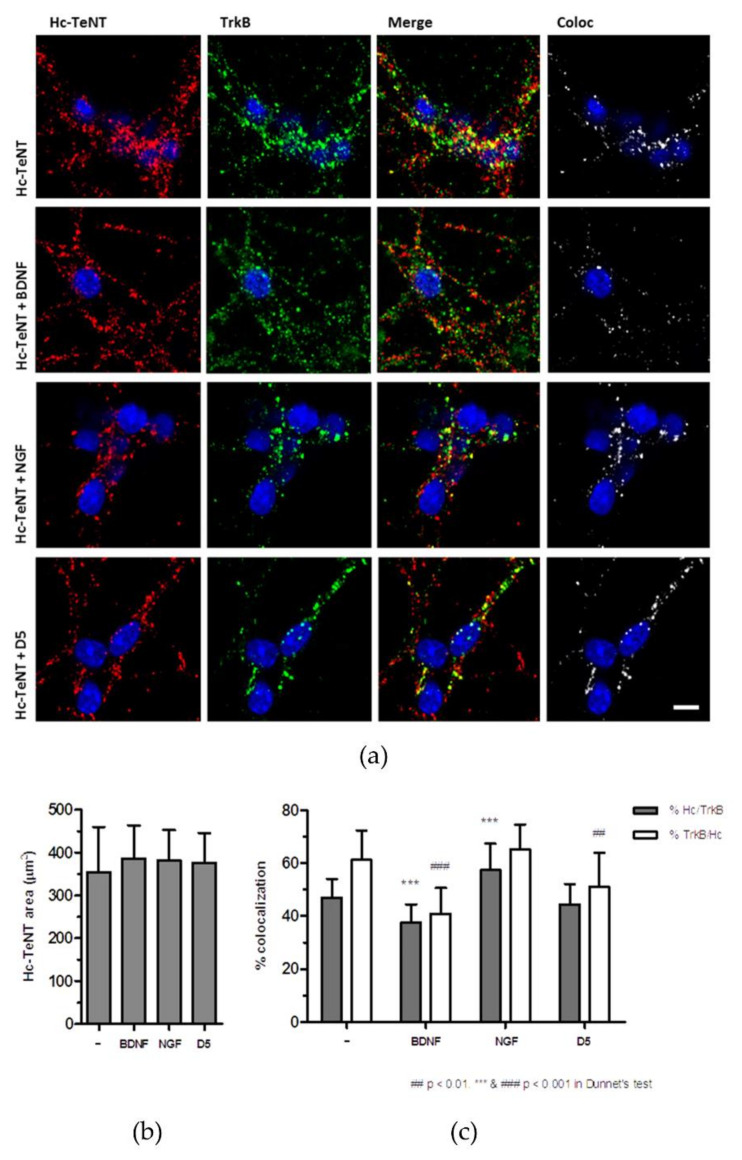
Hc-A555 uptake competition with BDNF, NGF, and the D5 peptide. (**a**) Cerebellar granule neurons were/was incubated with 20 nM Hc-A555 in the presence of BDNF (30 nM), NGF (30 nM), or the D5 peptide (20 μM), in internalization conditions. Cells were fixed and immunostained to detect TrkB. Images were obtained in/using a confocal microscope (scale bar 5 μm) and analyzed with Imaris software to obtain colocalizing points (right panel). (**b**) Mean on the area occupied by Hc-A555 signal in each condition. (**c**) Colocalization percentage between Hc-TeNT and TrkB calculated in each condition. Mean ± SD of at least 20 fields are shown, with one-way ANOVA and Dunnet’s *post hoc* test (*** *p* < 0.001 compared to % Hc/TrkB in non-treated, ## *p* < 0.01 and ### *p* < 0.001 compared to % TrkB/Hc in non-treated).

**Figure 5 molecules-26-03988-f005:**
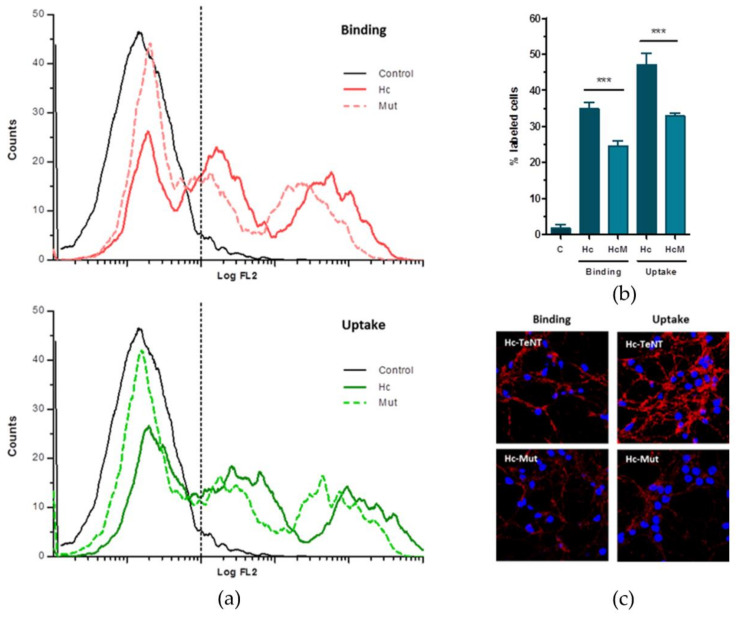
TrkB and Hc-TeNT associates in mouse NMJ and spinal motoneurons. The three panels show the following: (**a**) Upper part with a histogram of flow cytometry analysis with Hc-Mut in comparison to membrane-bound Hc-TeNT, as well as the endocyted situation in the lower part; (**b**) internalization assays; and (**c**) analysis of binding and internalization by confocal microscopy. *** *p* < 0.001 using one-way ANOVA followed by Bonferroni’s *post hoc* test.

**Figure 6 molecules-26-03988-f006:**
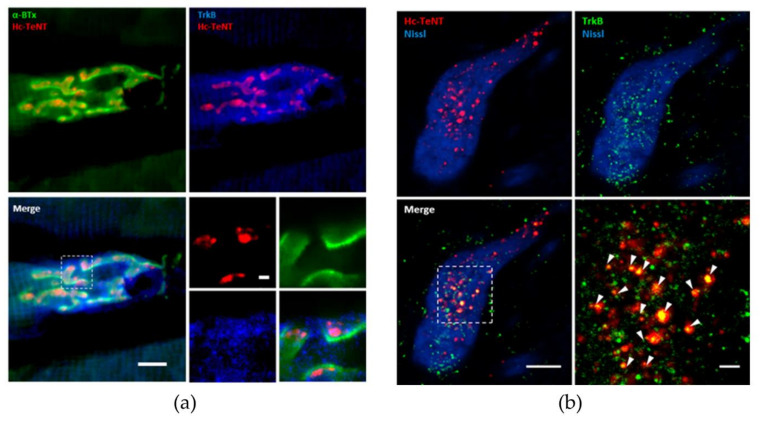
Localization of Hc-A555 after intramuscular injection in mice. (**a**) Neuromuscular junction in a tibialis anterior muscle slice, 24 h after injection. Postsynaptic membranes were stained with α-BTx in green and TrkB receptor in blue (Scale bar = 10 μm). Amplification is shown in the right bottom panel (Scale bar = 1 μm). (**b**) Motoneuron soma in the lumbar region of the spinal cord innervating the tibialis anterior muscle. Somas were visualized by Nissl staining (blue), and an antibody against TrkB was used (green) (scale bar = 10 μm). Amplification in the right bottom panel shows a large number of vesicles positives for Hc-A555 and TrkB (scale bar = 2 μm).

**Figure 7 molecules-26-03988-f007:**
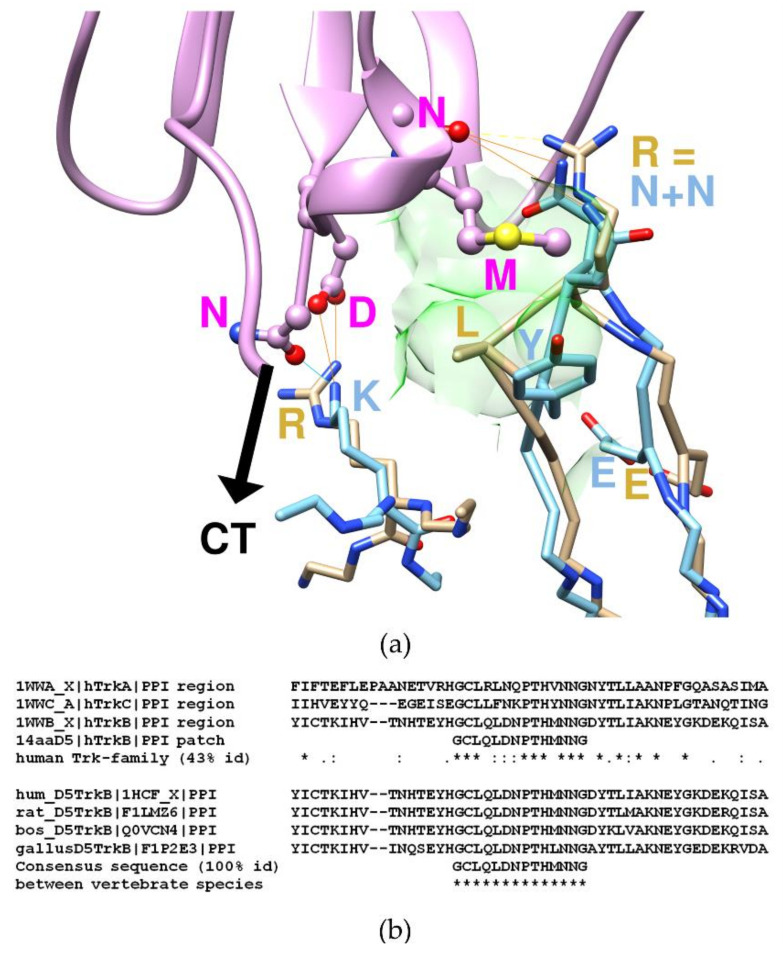
3D- and 2D-alignments of structures and sequences. (**a**) Detailed view of the binding patch between computed (TeNT, blue), and observed (neurotrophin, beige) complexes. To this end, the crystal complex (PDB code: 1HCF) between the neurotrophin (NT) ligand (beige) and the neurotrophin receptor (NTR) ectodomain D5 of TrkB was superpositioned on the proposed complex between Hc-TeNT (blue) and D5 (magenta). A hydrophobic zone forms part of the binding patch and is displayed by its contouring atomic surfaces (green). (**b**) Multiple segment alignment studies of the binding patch segment and its neighboring region. The first block is to demonstrate how closely related the three members of the neurotrophin receptor family Trk are in the segment of 14 residues and its nearby region. The second block shows how well conserved the binding path segment is among vertebrate organisms: human (h, hum), rattus (rat), bos (cow), and gallus (chicken). Symbols: asterisk * marks identity, semicolon: marks homology, blank space marks nonconserved position. A hyphen (-) is a gap (amino acid deletion). Of note, the atom IDs are those given in the PDB entries 1HCF [12] and 1A8D [27].

**Figure 8 molecules-26-03988-f008:**
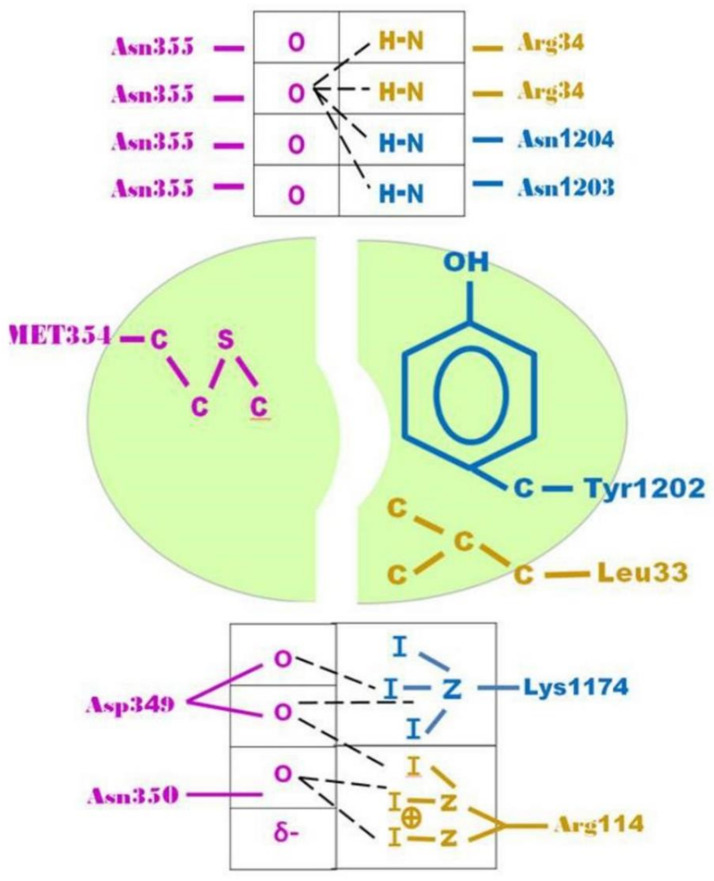
Schematic view of the binding analogy for TeNT and neurotrophin 4/5 ligands to NTR [12]. The hotspot is dissected into receptor of TrkB (to the left, magenta) and the two ligands (to the right, blue: Hcc fragment and beige: NT4/5). Equivalent amino acids are boxed. They coincide in space and function by evolutionary convergence to bind to d5. Sequence and structure alignments show that there is no homology and no common fold units.

## Data Availability

All data can be send by email on request.

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
