# Peer review of "Interaction between a Novel Oligopeptide Fragment of the Human Neurotrophin Receptor TrkB Ectodomain D5 and the C-Terminal Fragment of Tetanus Neurotoxin"

_molecules, 2021, doi:10.3390/molecules26133988_

Round 1

Reviewer 1 Report

In this work the authors used in silico and biochemical approaches to investigate the interaction between the ectodomain D5 of TrkB receptors and the C-terminal region of the tetanus neurotoxin. The results indicate that the interaction of the tetanus toxin with TrkB receptors mediate the internalization of the toxin. This is a novel contribution in the field, but unfortunately there are results presented in some of the figures that are not described in the results section and there are results described in this section that are not illustrated in the figures provided. Therefore, at this point it is difficult to assess some of the data.

Major comments:

  1. Figures 1 and 2 are not described in the results section. The cell type used in the experiments reported in Figure 1 should be indicated in the figure caption.
  2. Figure 1C: The representative results of the western blot experiments suggest that the peptide Hc-TeNT activate TrkB receptors, as determined with a phosphospecific antibody. However, at this point the results obtained are not statistically significant. In order to be able claim that the peptide activates TrkB receptors (as stated in the discussion – page 16, line 458) additional experiments should be performed. Which is the hypothesis to explain the activation of the receptors by the peptide? In the figure caption it is stated that 5 independent experiments were performed. Were these experiments performed in distinct preparations? This should also be clarified. The same applies to the experiments described in the other figures.
  3. Page 8 (first paragraph): the authors state that ‘... the quantity of endocytosed Hc was not affected by the addition of competitor molecules (Figure 4b)’. However, Figure 4b reports the area of Hc-TeNT, which does not give information about the intensity of the signal.
  4. Page 8 (line 242): It is concluded that NGF may facilitate the internalization of Hc and TrkB together. This is surprising because TrkB receptors do not bind NGF and as stated in page 7 (line 219) the CGN do not express TrkA receptors for NGF. Which is the hypothesis to explain these results?
  5. Page 10 (line 268): The results of Figure 1b do not illustrate this claim.
  6. Page 10 (last paragraph): The results described in this paragraph are not illustrated in the figures provided (certainly not in Figure 5, as indicated in this paragraph).
  7. Page 11 (line 303): SV2 staining is not shown in Figure 6a, in contrast with the information provided here.
  8. Figure 6: This figure is not very informative since no quantifications were provided and, as pointed out above, it is not indicated how many independent experiments were performed, using different preparations.

Minor comments:

  1. At several points in the manuscript the authors described the temperature used as 37ºC instead of 37°C. This should be corrected.
  2. The manuscript requires editing for typos. The last sentence of page 17 should be revised.
  3. Page 18 (last paragraph of the discussion): The authors may wish to point out that PC12 cells express TrkA receptors, since it makes their point stronger.

Author Response

Please, see attached PDF. Thank you. 

Reviewer 2 Report

Major comments:

The authors systematically teased apart the mechanism by which the Tetanus neurotoxin enters motor neurons. By combining in silico, in vitro, and in vivo work, they were able to collect compounding evidence pointing to a specific family of receptors responsible for cellular update of the toxin.

Overall, I find the study to be a very good one with results that have important implications in the advancement of botulism toxicity. However, I found the manuscript difficult to read and work through due to lack of organization and a clear writing. The figures are not presented in the same order that they’re mentioned in the text and as a result I found myself spending a lot of time trying to work out what they meant. The manuscript is also difficult to read due to grammatical problems that fog up the important points. I pointed out a few in the line comments below but can’t list everything. Several more rounds of vigorous proofreading and refinement would significantly improve this paper.

Specific comments:

  • L20 – the first sentence of the abstract is confusing. Perhaps delete the first part before the comma and just start with “We studied..”
  • L54 – remove “a”
  • L55 – change “Since the last century’s decade of the 80s” to “Since the 80s, “
  • L58 – move the parenthesized description “(cell uptake)” to L54 when the term internalization is first mentioned.
  • L94 – “polysialogangliosides” should be singular here
  • L105-107 – this sentence technically isn’t a question so there shouldn’t be a question mark at the end.
  • L118 – what is “mayor tenet”?
  • L183 - The end of the introduction needs a summary of the aims of the study. It’s not clear after reading the background what the main goals are.
  • L187 – it seems out of place to have a single sentence on its own here. Why not just move this to the methods section instead?
  • Figure 1 panel B – what is H4? Also, instead of “peptide” it would be clearer to use “ug protein” or “ug peptide.” Also, in the legend, rather than describing the methods in detail, please describe what the blot is telling us.
  • Figure 1 panel C – what is NT? Also, again, please describe more clearly what the western blot and graph are showing us/confirming.
  • Figure 1 isn’t mentioned in the text until several pages down – why is it the first figure? Also, figure 2 isn’t mentioned at all in the results section. I suggest either making these supplemental figures or mention these results at the beginning of the results section and keep the figures where they are.
  • Shouldn’t figure 3 be mentioned in section 2.1 of the results?
  • There are many abbreviations being used throughout the manuscript and it would help readers to have a table of definitions so they can quickly refer to it when they can’t keep them memorized in their heads.
  • Figure 3A –What does the decrease in blue and red staining with increased competitive molecule concentration show? Mention this in the legend.
  • L264 – alanine should not have an apostrophe. Also were all three residues substituted for A (Y266A, K311A, and E343A)?
  • L266 – is the reference to Figure 1B correct here? I don’t see any indication of mutations in this dot blot figure.
  • L268 – sometimes in vitro is italicized, sometimes not. It doesn’t need to be italicized.
  • L353 – residue formatting isn’t consistent as above, on line 264
  • L439 – change “has” to “have”

Author Response

Please, find the reply in attached PDF.
